

# Towards near-real time air pollutant and greenhouse gas emissions: lessons learned from multiple estimates during the COVID-19 Pandemic

Marc Guevara[1], Hervé Petetin[1], Oriol Jorba[1], Hugo Denier van der Gon[2], Jeroen Kuenen[2], Ingrid Super[2], Claire Granier[3,4], Thierno Doumbia[3], Philippe Ciais[5], Zhu Liu[6], Robin D. Lamboll[7], Sabine Schindlbacher,[8] Bradley Matthews[8] and Carlos Pérez Garcia-Pando[1,9]

[1] Barcelona Supercomputing Center, Barcelona, Spain
[2] TNO, Department of Climate, Air and Sustainability, Utrecht, the Netherland
[3] Laboratoire d'Aérologie, Université de Toulouse, CNRS/UPS, Toulouse, France
[4] NOAA Chemical Sciences Laboratory–CIRES/University of Colorado, Boulder, CO, USA
[5] Laboratoire des Sciences du Climat et de l'Environnement (LSCE/IPSL), CEA-CNRS-UVSQ, Univ Paris-Saclay, Gif-sur-Yvette, France
[6] Department of Earth System Science, Tsinghua University, Beijing, 100084, China
[7] Grantham Institute for Climate Change and the Environment, Imperial College London, London, UK
[8] Environnent Agency Austria, Spittelauer Lände 5, 1090 Vienna, Austria
[9] ICREA, Catalan Institution for Research and Advanced Studies, Barcelona, Spain

*Correspondence to*: Marc Guevara (marc.guevara@bsc.es)

**Abstract.** The 2020 COVID-19 crisis caused an unprecedented drop in anthropogenic emissions of air pollutants and

greenhouse gases. Given that emissions estimates from official national inventories for the year 2020 were not reported until two years later, new and non-traditional datasets to estimate near-real time emissions became particularly relevant and widely used in international monitoring and modelling activities during the pandemic. This study investigates the impact of the COVID-19 pandemic on 2020 European (the 27 EU Member States and the UK) emissions by comparing a selection of such near-real time emission estimates, with the official inventories that were subsequently reported in 2022 under the Convention

on Long-Range Transboundary Air Pollution (CLRTAP) and the United Nations Framework Convention on Climate Change (UNFCCC). Results indicate that annual changes in total 2020 emissions reported by official and near-real time estimates are fairly in line for most of the chemical species, with $NO_x$ and fossil fuel $CO_2$ being reported as the ones that experienced the largest reduction in Europe in all cases. However, large discrepancies arise between the official and non-official datasets when comparing annual results at the sector and country level, indicating that caution should be exercised when estimating changes

in emissions using specific near-real time activity datasets, such as time mobility data derived from smartphones. Main examples of these differences are observed for manufacturing industry $NO_x$ (relative changes ranging between -21.4% and -5.4%) and road transport $CO_2$ (relative changes ranging between -29.3% and 5.6%) total European emissions. Additionally, significant discrepancies are observed between the quarterly and monthly distribution of emissions drops reported by the various near-real time inventories, with differences up to a factor of 1.5 for total $NO_x$ during April 2020, when restrictions

were at their maximum. For residential combustion, shipping and public energy industry, results indicate that changes in emissions that occurred between 2019 and 2020 were mainly dominated by non-COVID-19 factors including meteorology,



the implementation of the Global Sulphur Cap and the shutdown of coal-fired power plants as part of national decarbonization efforts, respectively. The potential increase in NMVOC emissions from the intensive use of personal protective equipment such as hand sanitizer gels is considered in a heterogeneous way across countries in official reported inventories, indicating

the need for some countries to base their calculations on more advanced methods. The findings of this study can be used to better understand the uncertainties of near-real time emissions and how such emissions could be used in the future to provide timely updates to emission datasets that are critical for modelling and monitoring applications.

## 1    Introduction

Under the United Nations Economic Commission for Europe (UNECE) Convention on Long-Range Transboundary Air

Pollution (CLRTAP) (UNECE, 2012) and the United Nations Framework Convention on Climate Change (UNFCCC) (UNFCCC, 1992), as well as corresponding EU legislation (i.e., the National Emission reduction Commitments Directive, European Commission, 2016; the Monitoring Mechanism Regulation, European Commission, 2013), EU Member States and the UK are obliged to report annual emission inventories of air pollutants (AP) and greenhouse gases (GHG). These reported inventories form the basis for monitoring progress towards collective goals as well as national emission ceilings and reduction

commitments (e.g., the Effort Sharing Decision, European Commission, 2009). Parties must submit their emission inventories on an annual basis in accordance with the corresponding reporting guidelines and following the emission estimation methodologies described in the European Monitoring and Evaluation Programme/ European Environment Agency (EMEP/EEA) air pollutant emission inventory guidebook (EEA, 2019) for AP and the Intergovernmental Panel on Climate Change (IPCC) Guidelines for National Greenhouse Gas Inventories (IPCC, 2019) for GHG.


Despite providing consistent and robust time series of emission estimates, official emission inventory submissions are reported with a two-year time lag. The lagged reporting deadlines (i.e., reporting in a given year (Y) shall typically include annual emissions estimates from 1990 to Y-2) reflect the time needed to finalize accurate national statistics (e.g., official energy consumption statistics) and the cost, time and effort entailed to collect and process them for compiling emission inventories.

As a result, in addition to the inherent uncertainties of emission inventories, this time lag can introduce additional uncertainties when these datasets are used (and extrapolated) in certain modelling applications, mainly air quality forecasting systems, as they may not represent current emission sources accurately (Tong et al., 2012). This limitation can be largely amplified in the event of major and unexpected emission changes, such as during the 2008–2009 global economic recession (Castellanos and Folkert, 2012; Peters et al., 2012) or more recently the COVID-19 pandemic. Given the sharp drop in mobility and associated

emissions caused by the COVID-19 crisis, alternative methods to estimate near-real time emissions were developed, with the objective of contributing to numerical modelling exercises aiming at understanding the impact of those emission changes on air quality levels (e.g., Badia et al., 2021; Barré et al., 2021; Gaubert et al., 2021; Schneider et al., 2021). These emission datasets included the use of near-real time activity information that is not traditionally used in official reported inventories,



such as mobility data derived from smartphones, congestion statistics obtained from navigation applications or near-real time
electricity load and generation statistics published by national transmission system operators, among others. Seminal studies
tackling the impact of COVID-19 upon primary emissions include Le Quéré et al. (2020), Forster et al. (2020), Guevara et al.
(2020 and 2022), Liu et al. (2020a and 2020b), Doumbia et al. (2021), Harkins et al. (2021) and Zheng et al. (2021). In reaction
to the COVID-19 pandemic and the growing interest in near-real time emission estimates, some European countries started to
publish quarterly and monthly estimates of emissions based on preliminary energy data (e.g., CITEPA, 2022). Results from
these studies suggest that near-real time emission estimates could be used to provide timely updates to emission trends,
especially in the case of other significant and unexpected anthropogenic emission changes (e.g., economic and energy crisis,
armed conflicts). However, before they can be used to complement official emission inventories and be integrated into air
quality forecasting systems, an assessment of their reliability and associated uncertainty is needed.

This study provides an intercomparison of 2020 emission changes derived from official reported inventories and multiple near-
real time estimates for various AP (i.e., $NO_x$, NMVOC, $SO_2$, $NH_3$, PM10, PM2.5) and GHG (i.e., $CO_2$ and $CH_4$) for European
Union countries (EU27) plus United Kingdom (EU27 + UK). Specifically, we evaluated the magnitude of relative emission
changes reported by both official and non-official estimates for individual pollutant sources. Considering the emission drops
associated with COVID-19 restrictions occurred in a heterogeneous way and at specific periods of the year, the study not only
focuses on annual emission changes, but also includes comparisons of intra-annual variability reported by the different near-
real time emission estimates (i.e., quarterly and monthly level). The results of this inter-comparison exercise are used to
produce recommendations on how best to approach near-real time emission estimates.

Section 2 describes the methods and datasets considered for the intercomparison, while Section 3 discusses the results obtained
in terms of annual, quarterly and monthly relative emission changes by pollutant, country and sector. The main conclusions
and lessons learned from this work are provided in Section 4.



## 2    Methodology

This study compares AP and GHG emission changes in 2020 as provided by 4 near-real time emission estimates and the
official reported inventories under the CLRTAP and UNFCCC, respectively. A description of each dataset and summary of
methodologies is provided in Table 1 and sections 2.1 and 2.2.

**Table 1 Main characteristics of the emission datasets considered in the intercomparison work**

| Dataset | Type of data | Spatial coverage (resolution) | Temporal coverage (resolution) | Species | Sectors | Reference |
|---|---|---|---|---|---|---|
| guevaraetal | Relative adjustment factors | Europe (country level) | Jan-Dec 2020 (daily, weekly[*]) | AP and GHG [***] | All GNFR sectors except for J_Waste, K_AgriLivestock and L_AgriOther | Guevara et al. (2020 and 2022) |
| doumbiaetal | Relative adjustment factors | Global (country level) | Jan-Dec 2020 (daily/monthly[**]) | AP and GHG [***] | Road transport, aviation shipping, power, industry, residential/commercial | Doumbia et al. (2021) |
| forsteretal | Relative adjustment factors | Global (country level) | Jan-Dec 2020 (daily) | AP and GHG [****] | Road transport, residential, power, industry and aviation | Forster et al. (2020) |
| liuetal | Absolute emissions | Global (country level for EU27 plus UK) | Jan 2019 until present day | $CO_2$ | Road transport, residential, power, industry, aviation, shipping | Liu et al. (2020a, 2020b) |
| emep_ceip | Absolute emissions | Member states under the CLRTAP (country level) | 1990-2020 (annual) | NOx, SOx, CO, NMVOC, $NH_3$, PM10, PM2.5 | All GNFR sectors | CEIP (2022) |
| unfccc | Absolute emissions | Member states under the UNFCCC (country level) | 1990-2020 (annual) | $CO_2$, $CH_4$, $N_2O$ | All CRF sectors | UNFCCC (2022) |

[*] Daily for all sectors except for shipping (weekly)
[**] Daily for all sectors except for shipping and aviation (monthly)
[***] Relative adjustment factors are country-, sector- and species-dependent
[****] Relative adjustment factors are only country and sector-dependent (same factors assumed for all AP and GHG species)

## 2.1    Officially reported emissions

Officially reported emissions of NOx, NMVOC, CO, $SO_2$, $NH_3$ and PM2.5 (reporting year 2022) for 2019 and 2020 were
obtained from the EMEP Centre on Emission Inventories and Projections (CEIP, 2022; hereinafter referred to as emep_ceip),
containing sectoral emissions following the Gridded Nomenclature For Reporting (GNFR) classification system. Officially
reported 2019 and 2020 emissions of $CO_2$ and $CH_4$ (reporting year 2022) were obtained from the national inventory
submissions to the UNFCCC (UNFCCC, 2022; hereinafter referred to as unfccc). GHGs emission data at the Common
Reporting Format (CRF) level was converted to the GNFR classification system according to the CRF-GNFR crosswalk of
Kuenen et al. (2022a). A detailed description of the activity data and emission factors used to estimate official reported
emissions is provided by each country under the Informative Inventory Reports (IIRs) for the AP (CEIP, 2022) and the National
Inventory Reports (NIR) for the GHG (UNFCCC, 2022).



## 2.2 Near-real time estimates

Methodologies and proxies used by each near-real time database to derive emission estimates are summarised in Table 2.

The Copernicus Atmosphere Monitoring Service (CAMS) COVID-19 emission adjustment factors dataset (Guevara et al., 2022; hereinafter referred to as guevaraetal) is a European dataset of daily sector-, pollutant- and country-dependent emission adjustment factors associated with the COVID-19 mobility restrictions for the year 2020. Adjustment factors are expressed as

a percentage of emission changes compared to a 2020 business-as-usual scenario, i.e., the emissions that would have been released in 2020 in the absence of COVID-19 restrictions and under the same meteorological conditions. The resulting dataset covers a total of nine emission sectors, which are grouped according to the GNFR classification system, including road transport, energy industry, manufacturing industry, residential and commercial combustion, aviation, shipping, off-road transport, use of solvents, and fugitive emissions from transportation and distribution of fossil fuels. The adjustment factors

were developed considering activity information traditionally used to estimate emissions, such as energy statistics or traffic counts, as well as information derived from Google COVID-19 Community Mobility data (Google LLC, 2021) and machine learning techniques. The adjustment factors developed by Guevara et al. (2022) are pollutant-dependent and consider the heterogeneous impact of the COVID-19 restrictions across the different activities in some sectors (e.g., light-duty vehicles versus heavy-duty vehicles in the road transport sector, GNFR_F, or essential versus non-essential industrial activities in the

manufacturing industry sector, GNFR_B).

The COvid-19 adjustmeNt Factors fOR eMissions (CONFORM, Doumbia et al, 2021; hereinafter referred to as doumbiaetal) provides a global dataset of emission adjustment factors per country and sector that quantify relative changes in emissions compared to a business-as-usual situation in 2020. The activity factors are estimated using data from a variety of sources,

including google mobility reports for the road transport, residential and commercial and manufacturing industry sectors, total electricity load for the power generation sector, data on air transportation published by the Knowledge Center on Migration and Demography (KCMD) Dynamic Data Hub for the aviation sector and statistics on container ship port calls reported by the United Nations Conference on Trade and Development for maritime emissions. In contrast to the CAMS datasets, for each sector the constructed adjustment factors are homogeneous across species.


Like CONFORM, Forster et al. (2020) report a dataset of global emission adjustment factors that vary per country and sector and cover the whole year 2020 (hereinafter referred to as forsteretal). Adjustment factors are also derived in large part from Google mobility data (i.e., for the surface transport, residential, public and commercial and manufacturing industry sectors). For the power sector, weighted Google mobility data reported for the workplace, residential and retail categories are considered

to construct the adjustment factors, which are then scaled to match the $CO_2$ global emission change reported by Le Quéré et al. (2020). For the air traffic and maritime sector, the Le Quéré et al. (2020) emission trends for international and national



aviation and shipping are directly used. The developed adjustment factors were later used by Lamboll et al. (2021) to produce gridded projections of emission scenarios and run general circulation models to investigate the impact of national lockdown measures on climate. This slightly modified the approach to aviation emissions, which were globally scaled in proportion to

the total number of aircraft flying at that time, reported by FlightRadar24.

The Carbon Monitor initiative (Liu et al., 2020a and 2020b; hereinafter referred to as liuetal) provides estimates of global daily $CO_2$ emissions from fossil fuel combustion and cement production. Daily emissions are estimated from annual emissions from the Emissions Database for Global Atmospheric Research inventory (EDGAR, Crippa et al., 2019) in the base year 2019 and

a diverse range of activity data, which are used to downscale and extrapolate in time annual emissions to daily level from each sector. The activity proxies considered include electrical power generation, production data and production indices of industry processes, mobility data and mobility indices of road transportation (TomTom data for > 200 cities in Europe aggregated to country scale) and flight location data (FlightRadar24 database) and shipping mobility statistics for aviation and maritime transportation. Residential emissions are assumed to vary only according to population weighted daily temperature. Emissions

are reported per sector (for 6 sectors) and country or group of countries. A specific European version of Carbon Monitor was recently released, which reports emissions from each of the individual countries of the EU27 + UK bloc (Ke et al., 2022). The sectors included in the dataset are road transport, energy industry, manufacturing industry, residential and commercial buildings fuel use, aviation and shipping. Unlike the previous three datasets, Carbon monitor does not provide information on relative emission changes but estimates of daily absolute emissions from January 2019 until the present, with a ~ 3 months

latency after the time of emission.



**Table 2 Summary of the methodologies and proxies considered in the near-real time estimates per sector**

| Dataset | Energy industry | Residential/commercial combustion | Manufacturing industry | Road transport | Air traffic | Shipping |
|---|---|---|---|---|---|---|
| guevaraetal | Temperature corrected electricity demand data from ENTSO-E [1] using population-weighted ERA5 2-m ambient air temperature [2] | Google COVID-19 Mobility data [3] (average of retail and recreation, residential, and workplaces categories) adjusted with measured residential and commercial energy consumption statistics | Industrial production indexes from Eurostat [4] | Google COVID-19 Mobility data (transit stations category) adjusted with measured traffic counts | Airport movement statistics from EUROCONTROL [7] | CO2 AIS-based shipping emissions from STEAM (Jalkanen et al. 2016) |
| doumbiaetal | Electricity demand data from ENTSO-E [2] | Google COVID-19 Mobility data [3] (residential category) | Google COVID-19 Mobility data [3] (workplaces categories) | Google COVID-19 Mobility data [3] (transit stations category) | Official Aviation Guide measurements [8] in conjunction with data by the Knowledge Center on Migration and Demography Dynamic Data Hub [9] | Container ship port calls reported by the United Nations Conference on Trade and Development [11] |
| forsteretal | Google COVID-19 Mobility data [3] (average of retail and recreation, residential, and workplaces categories) | Google COVID-19 Mobility data (residential and retail and recreation categories) | Google COVID-19 Mobility data [3] (workplaces categories) | Google COVID-19 Mobility data [3] (transit stations category) | Relative emission changes reported by Le Quéré et al. (2020) | Relative emission changes reported by Le Quéré et al. (2020) |
| liuetal | Electricity generation data by production types from ENTSO-E [1] | Population-weighted heating degree days assuming no direct effect of COVID and other factors | Industrial production indexes from Eurostat [4] | TomTom congestion data [5] calibrated against car flux data (Paris) [6] | Individual commercial flights tracked by Flightradar24 [10] | Decline ratio reported by news reports [12] |

[1] ENTSO-E (2022)
[2] C3S (2017)
[3] Google LCC (2021)
[4] Eurostat (2021)
[5] https://www.tomtom.com/en_gb/traffic-index/, last access: November 2022
[6] https://opendata.paris.fr/pages/home/, last access: November 2022
[7] EUROCONTROL (2021)
[8] https://www.oag.com/coronavirus-airline-schedules-data, last access: November 2022
[9] https://migration-demography-tools.jrc.ec.europa.eu/data-hub/, last access: November 2022
[10] https://www.flightradar24.com, last access: November 2022
[11] https://unctad.org/news/covid-19-shipping-data-hints-some-recovery-global-trade, last access: November 2022
[12] Kinsey (2020)



### 2.3     Baseline for the estimation of 2020 relative emission changes

The estimation of relative changes ($RC_{s,c,p}$) in 2020 emissions per GNFR sector $s$, country $c$ and pollutant $p$ is computed as indicated in Eq.1:

$$RC_{s,c,p} = \left( \frac{Emis2020_{s,c,p} - EmisBaseline_{s,c,p}}{EmisBaseline_{s,c,p}} \right) * 100 \qquad \text{Eq. 1}$$

Where $Emis2020_{s,c,p}$ are the annual emissions reported for 2020 per GNFR sector $s$, country $c$ and pollutant $p$ and
$EmisBaseline_{s,c,p}$ are the annual emissions reported for the baseline scenario per GNFR sector $s$, country $c$ and pollutant $p$.

The baseline considered for the official reported emissions (i.e., emep_ceip and unfccc) and liuetal is the year 2019 (from EDGARv4.3 in liuetal) because the three datasets report emissions for that year as well as for 2020. For the three other near-real time datasets (i.e., guevaraetal, doumbiaetla, forsteretal) the baseline considered is the Copernicus CAMS-REG_v5.1 2020
business-as-usual (BAU) emission inventory (Kuenen et al., 2022b), which reports AP and GHG emissions for 2020 ignoring the impact of COVID-19, while 2020 emissions are estimated by combining this inventory with the adjustment factors reported by each dataset.

The use of different baselines implies that the relative changes estimated by official reported emissions and liuetal are not only
related to the effect of the COVID-19 restrictions, but also to other factors such as changes in meteorology or the implementation of new emission regulations between 2019 and 2020, while the computed relative changes for guevaraetal, doumbiaetla and forsteretal only account for the COVID-19 effect. Consequently, this comparison brings the opportunity of disentangling the COVID-19 impacts from other effects on 2020 emissions.

### 3     Comparison of changes in 2020 emissions

In this section, we compare relative changes in 2020 emissions as reported by the official and non-official estimates described in Sect. 2. The comparison focuses on EU27 + UK and is performed at the annual (Sect. 3.1) and monthly (Sect. 3.2) scale.

### 3.1     Annual emission changes

Annual changes in $NO_x$, NMVOC, CO, $SO_2$, $PM_{2.5}$, $NH_3$, $CO_2$ and $CH_4$ emissions per GNFR sector at the EU27+UK level are summarised in Table 3 and Table 4. Note that for the shipping sector we only show guevaraetal results because the other
datasets report emission changes for global shipping emissions and do not distinguish between European and non-European



sea regions. Figure 1 shows the relative contribution [%] of each GNFR sector to total 2019 emissions at the EU27 + UK level to support the analysis performed.

CO$_2$ (-12.2%) and NO$_x$ (-11.3%) are the pollutants that experienced the largest reduction in Europe according to official
estimates (unfccc and emep_ceip, respectively), the values reported by doumbiaetla and liuetal for CO$_2$ (-12.2% and -11.6%, respectively) and guevaraetal for NO$_x$ (-10.5%) being the ones closer to them. These findings are in line with the fact that road transport contributes substantially to CO$_2$ and NO$_x$ emissions (Fig. 1) and, at the same time, was the most affected sector by the COVID-19 restrictions, after aviation. Also, in agreement with official estimates, NH$_3$ and CH$_4$ are reported by the near-real time datasets as the species that experienced the lowest reductions (i.e., -1.1% and -1.4% according to official estimates
and between -0.9% and 0.1% according to non-official estimates). Considering that agricultural and waste management practices contribute to more than 80% of total NH$_3$ and CH$_4$ emissions (Fig. 1), the results reinforce the hypothesis that these activities remained mostly unaffected during the COVID-19 mobility restrictions and lockdowns.

For SO$_2$, CO and PM2.5, official relative emission changes reported by emep_ceip (-10.8%, -8.2% and -4.1%) are much larger
than the ones reported by guevaraetal (-4.6%, -4.7% and -2.1%). For SO$_2$, discrepancies between results are mainly driven by the differences reported for the public power sector (A_PublicPower), which represent more than 30% of total SO$_2$ (Fig. 1). For this sector, the three non-official estimates report changes in emissions ranging from -7.2% to -2.7%, which are significantly lower than the official estimates (-19.5%) (see Sect. 3.1.1 for further details). In the case of doumbiaetal and forsteretal, the underestimation in the public power sector is compensated by a significant overestimation of the SO$_2$ emission
reduction in the manufacturing industry sector (B_Industry). While official estimates report a reduction of 7.8%, doumbiaetal and forsteretal indicate reduction of 20.2 and 22.8%, respectively. A similar situation is observed for CO$_2$, for which only liuetal is in line with the emission changes reported by unfccc for the public power sector (i.e., -14.4% versus -11.8%).

For CO and PM2.5, differences in relative emission changes reported by guevaraetal and official estimates are mainly driven
by the discrepancies observed in the residential/commercial stationary combustion activities (C_OtherStaComb), the largest contributor to total emissions for these two species (Fig. 1). Guevaraetal shows an increase in emissions (1.7% for CO and 1.8% for PM2.5), while emep_ceip indicates a reduction of 2.7% for CO and 1.9% for PM2.5. The discrepancies are much larger when looking at the results reported by forsteretal (6.6% for both CO and PM2.5) and doumbiaetal (5.9% for CO and 6.0% for PM2.5). Sect. 3.1.3 goes into detail about the reasons for these discrepancies. As seen for SO$_2$, the good agreement
between CO and PM2.5 total emission changes reported by forsteretal/doumbiaetal and emep_ceip is the result of an error compensation: the aforementioned underestimation in the residential/commercial stationary combustion activities is balanced with an overestimation in the reductions reported by official estimates for the manufacturing industry (e.g., -3% according to emep_ceip versus -21.0% according to doumbiaetal for PM2.5) and road transport (e.g., -19% according to emep_ceip versus -27% according to forsteretal for CO) sectors.



It is worth mentioning the large reduction in $SO_2$ shipping emissions reported by official estimates (-46.3%), which is mainly
      caused by the 2020 Global Sulphur Cap, which entered into force on the first of January 2020. The reduction reported by
      guevaraetal for this sector and pollutant is much lower (-11%), as it only accounts for the impact of COVID-19 restrictions.
      Nevertheless, when looking at NOx and $CO_2$ shipping emissions, a better agreement is found between the relative changes
      reported by emep_ceip (-13.5%), unfccc (-11%) and guevaraetal (-11%), confirming that the larger reduction found for $SO_2$ is

mainly linked to a non-COVID-19 effect. Finally, for NMVOC guevaraetal reports the closest emission reduction value to
      official estimates, both being quite low (-2.1% and -2.5%, respectively).





**Table 3 Relative changes [%] in NOₓ, NMVOC, CO, SO₂, PM2.5 and NH₃ emissions per GNFR sector at the EU27+UK as reported by official (emep_ceip) and non-official (guevaraetal, doumbiaetal, forsteretal) estimates**

| GNFR | NOx | | | | NMVOC | | | |
|---|---|---|---|---|---|---|---|---|
| | emep_ceip | guevaraetal | doumbiaetal | forsteretal | emep_ceip | guevaraetal | doumbiaetal | forsteretal |
| A_PublicPower | -12.2 | -3.3 | -3.3 | -7.1 | -4.5 | -3.3 | -3.2 | -8.5 |
| B_Industry | -5.4 | -6.7 | -21.7 | -24.1 | -2.9 | -2.8 | -22.1 | -24.6 |
| C_OtherStaComb | -2.2 | -3.0 | -2.8 | -3.5 | -3.0 | 1.1 | 4.7 | 5.3 |
| D_Fugitive | -11.1 | -10.7 | 0.0 | 0.0 | -12.7 | -10.1 | 0.0 | 0.0 |
| E_Solvents | -17.0 | 0.0 | 0.0 | 0.0 | 2.1 | -1.3 | 0.0 | 0.0 |
| F_RoadTransport | -18.4 | -16.8 | -23.9 | -28.7 | -13.0 | -18.8 | -22.5 | -27.5 |
| G_Shipping | -13.5 | -11.0 | - | - | -12.2 | -11.0 | - | - |
| H_Aviation | -57.4 | -55.7 | -41.1 | -52.9 | -55.5 | -54.9 | -40.9 | -52.7 |
| I_Offroad | -7.3 | -1.7 | 0.0 | 0.0 | -5.2 | -2.0 | 0.0 | 0.0 |
| J_Waste | 0.7 | 0.0 | 0.0 | 0.0 | -10.3 | 0.0 | 0.0 | 0.0 |
| K_AgriLivestock | -0.3 | - | - | - | -0.4 | 0.0 | 0.0 | 0.0 |
| L_AgriOther | 1.4 | 0.0 | 0.0 | 0.0 | 0.3 | 0.0 | 0.0 | 0.0 |
| Total except G_Shipping | -11.3 | -10.5 | -15.9 | -19.3 | -2.1 | -2.5 | -2.8 | -3.3 |
| | CO | | | | SO₂ | | | |
| | emep_ceip | guevaraetal | doumbiaetal | forsteretal | emep_ceip | guevaraetal | doumbiaetal | forsteretal |
| A_PublicPower | -3.9 | -3.2 | -3.1 | -7.3 | -19.5 | -2.9 | -2.7 | -7.2 |
| B_Industry | -12.3 | -7.3 | -20.9 | -23.2 | -7.8 | -6.4 | -20.2 | -22.8 |
| C_OtherStaComb | -2.7 | 1.7 | 5.9 | 6.6 | -1.2 | -0.5 | 1.5 | 1.4 |
| D_Fugitive | -8.7 | -6.5 | 0.0 | 0.0 | -11.5 | -9.2 | 0.0 | 0.0 |
| E_Solvents | -8.5 | 0.0 | 0.0 | 0.0 | -19.9 | 0.0 | 0.0 | 0.0 |
| F_RoadTransport | -19.2 | -17.8 | -22.2 | -27.1 | -13.0 | -17.6 | -25.2 | -30.4 |
| G_Shipping | -7.4 | -11.0 | - | - | -46.3 | -11.0 | - | - |
| H_Aviation | -48.4 | -51.2 | -38.4 | -50.1 | -59.4 | -56.1 | -42.5 | -54.0 |
| I_Offroad | -2.4 | -2.7 | 0.0 | 0.0 | -17.2 | -0.3 | 0.0 | 0.0 |
| J_Waste | -1.4 | 0.0 | 0.0 | 0.0 | -2.6 | 0.0 | 0.0 | 0.0 |
| K_AgriLivestock | - | - | - | - | - | - | - | - |
| L_AgriOther | -3.8 | 0.0 | 0.0 | 0.0 | -3.9 | 0.0 | 0.0 | 0.0 |
| Total except G_Shipping | -8.2 | -4.7 | -6.4 | -7.6 | -10.8 | -4.6 | -9.4 | -12.1 |
| | NH₃ | | | | PM2.5 | | | |
| | emep_ceip | guevaraetal | doumbiaetal | forsteretal | emep_ceip | guevaraetal | doumbiaetal | forsteretal |
| A_PublicPower | 7.3 | -3.4 | -3.1 | -9.3 | -6.6 | -3.0 | -2.8 | -7.5 |
| B_Industry | 0.0 | -3.6 | -20.2 | -22.7 | -3.0 | -6.6 | -21.0 | -23.5 |
| C_OtherStaComb | -2.6 | 1.7 | 5.8 | 6.4 | -1.9 | 1.8 | 6.0 | 6.6 |
| D_Fugitive | -4.4 | -0.7 | 0.0 | 0.0 | -19.0 | -8.5 | 0.0 | 0.0 |
| E_Solvents | -6.1 | 0.0 | 0.0 | 0.0 | -18.8 | 0.0 | 0.0 | 0.0 |
| F_RoadTransport | -16.3 | -17.8 | -23.3 | -28.4 | -16.3 | -16.3 | -23.5 | -28.4 |
| G_Shipping | -6.2 | - | - | - | -19.4 | -11.0 | - | - |
| H_Aviation | -55.7 | -55.9 | -49.3 | -45.4 | -58.1 | -54.4 | -34.7 | -55.4 |
| I_Offroad | -3.5 | -1.2 | 0.0 | 0.0 | -8.2 | -1.9 | 0.0 | 0.0 |
| J_Waste | -0.4 | 0.0 | 0.0 | 0.0 | 1.4 | 0.0 | 0.0 | 0.0 |
| K_AgriLivestock | -0.1 | 0.0 | 0.0 | 0.0 | -0.5 | 0.0 | 0.0 | 0.0 |
| L_AgriOther | -1.5 | 0.0 | 0.0 | 0.0 | -2.6 | 0.0 | 0.0 | 0.0 |
| Total except G_Shipping | -1.1 | -0.2 | -0.5 | -0.5 | -4.1 | -2.1 | -2.8 | -3.6 |





**Table 4 Relative changes [%] in CO₂ and CH₄ emissions per GNFR sector at the EU27e+UK as reported by official (unfccc) and non-official (guevaraetal, doumbiaetal, forsteretal, liuetal) estimates**

| GNFR | CO₂ | | | | | CH₄ | | | |
|---|---|---|---|---|---|---|---|---|---|
| | unfccc | guevaraetal | doumbiaetal | forsteretal | liuetal | unfccc | guevaraetal | doumbiaetal | forsteretal |
| A_PublicPower | -14.4 | -3.4 | -3.2 | -7.2 | -11.8 | 0.4 | -3.0 | -3.1 | -6.4 |
| B_Industry | -6.7 | -6.6 | -20.7 | -23.1 | -7.7 | -6.6 | -4.9 | -21.0 | -23.5 |
| C_OtherStaComb | -1.3 | -1.5 | 0.1 | -0.1 | -2.0 | -1.0 | 1.2 | 4.6 | 5.1 |
| D_Fugitive | -12.5 | -4.3 | 0.0 | 0.0 | - | -5.2 | -6.7 | 0.0 | 0.0 |
| E_Solvents | -1.5 | 0.0 | 0.0 | 0.0 | - | -12.4 | 0.0 | 0.0 | 0.0 |
| F_RoadTransport | -13.8 | -16.5 | -24.4 | -29.3 | -5.6 | -13.9 | -18.7 | -23.1 | -28.0 |
| G_Shipping | -11.0 | -11.0 | - | - | - | 5.6 | - | - | - |
| H_Aviation | -57.6 | -56.0 | -41.7 | -52.9 | -58.2 | -57.4 | -56.1 | -43.4 | -53.2 |
| I_Offroad | -12.5 | -1.6 | 0.0 | 0.0 | - | -16.8 | -1.9 | 0.0 | 0.0 |
| J_Waste | -1.4 | 0.0 | 0.0 | 0.0 | - | -2.1 | 0.0 | 0.0 | 0.0 |
| K_AgriLivestock | - | - | - | - | - | -0.1 | 0.0 | 0.0 | 0.0 |
| L_AgriOther | - | - | 0.0 | - | - | 5.6 | 0.0 | 0.0 | 0.0 |
| Total except G_Shipping | -12.2 | -7.2 | -12.2 | -15.2 | -11.6 | -1.4 | -0.9 | -0.1 | -0.2 |

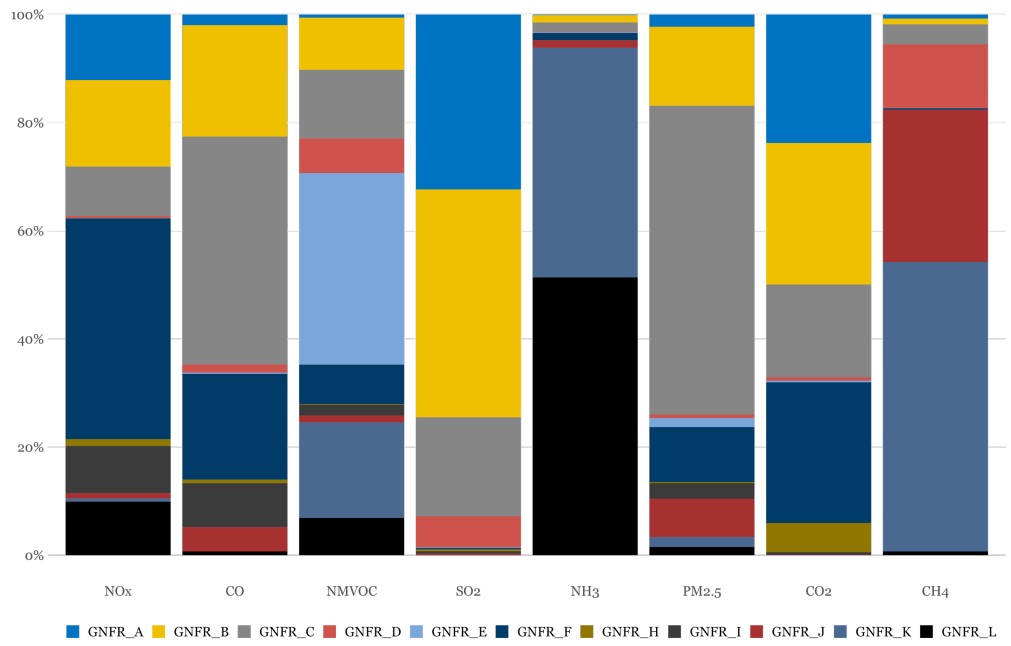

Figure 1: Sectoral contributions [%] to total 2019 emissions at the EU27 + UK level (CEIP, 2022; UNFCCC, 2022). Emissions are reported following the Gridded Nomenclature For Reporting (GNFR) classification system. Shipping emissions (GNFR_G) are excluded from the analysis.





Changes observed at the country level per individual sector are discussed in the following Sect. 3.1.1 to Sect. 3.1.8. For each

sector, we focus the analysis on the species to which the sector reports a contribution larger than 20% to the total EU27+UK bloc emissions of the respective species (Fig. 1). For those sectors with contributions lower than 20% for any species (i.e., aviation, fugitive emissions from fossil fuels, off-road mobile sources), we considered the most representative species. Note that sectors GNFR_J (waste management), GNFR_K (agriculture, livestock) and GNFR_L (agriculture, other practices including use of fertilizers and agricultural waste burning) were excluded from the discussion as all the near-real time datasets

assumed that emissions from these sources did not change in 2020 due to a lack of specific activity information or because of the nature of the European COVID-19 restrictions policies, which considered these activities to be essential during lockdowns. As shown in Fig. S1, this hypothesis is consistent with the official estimates, which report relative changes in emissions of maximum +/-5% in most countries.




### 3.1.1 Public power

Figure 2a and 2b shows the relative $SO_2$ and $CO_2$ emission changes [%] reported by each dataset per country and at the EU27+UK level for the public power sector (GNFR_A).

For $SO_2$, the three near-real time datasets consistently report much lower relative changes than official estimates. This discrepancy is partially because guevaraetal and doumbiaetal assume that COVID-19 restrictions had an impact on total electricity demand, but not on the electricity mix, which slightly shifted slightly towards renewables and therefore implied an additional reduction of activity in fossil fuel power plants (IEA, 2021). However, one of the most relevant aspects of these discrepancies is the role that national decarbonization trends played in the drop of emissions between 2019 and 2020. This is

illustrated with the examples of Spain and Estonia, which are among the countries showing the largest drop of emissions according to official reports (i.e., -60% and -45%, respectively). For Spain, the reduction is mainly related to the shutdown of seven coal-fired power plants in June 2020 as they were unable to comply with stricter EU air pollution standards (Europe Beyond Coal, 2022, Fig. 3), whereas the reduction in Estonia is due to a drop of 44% in the electricity produced by oil shale (IEA, 2022). In both countries, these reductions are part of commitments to a sustainable transition towards climate neutrality

and that were started to be executed before the outbreak of the COVID-19 crisis, e.g., in Estonia, power production from oil shale has already dropped by more than half from 9.5 TWh to 4.3 TWh between 2018 and 2019.

For $CO_2$, similar discrepancies are observed between official and near-real time estimates, except for liuetal, whose results are much more in line with unfccc because its relative emission changes consider changes in the electricity production by fuel type

between 2019 and 2020, and therefore integrate the impact of the decarbonization efforts in the electric power sector, even though emission factors for each fuel type are assumed to be constant (equal to the year) in their methodology. This is clearly observed in Spain and Estonia, where liuetal reproduces the official reported drops well (i.e., -26% according to unfccc and -21% according to liuetal for Spain; -35% according to unfccc and -40% according to liuetal for Estonia).

Changes in official emissions reported by emep_ceip and unfccc are generally consistent across $SO_2$ and $CO_2$, except in some countries such as Lithuania and Latvia, where $SO_2$ remains almost unchanged (4% and -6%) while $CO_2$ significantly increases (more than 50%) and decreases (-25%), respectively. For these two countries, the reason for this inconsistency is a significant change in the amount of electricity produced from natural gas between 2019 and 2020 (90% for Lithuania and -65% for Latvia; IEA, 2022), which had a significant impact in $CO_2$ emissions, but was almost negligible in terms of $SO_2$ changes due to the

low Sulphur content associated with this fuel. In other countries, such as Luxemburg and Croatia, the large discrepancies between changes in $SO_2$ (increases of approximately +50% and +100%, respectively) and $CO_2$ emissions (changes below 5%) may indicate an issue with the reported data.





### 3.1.2    Manufacturing industry

As shown in Fig. 2c and 2d, guevaraetal and liuetal are the near-real time estimates that present the closest values to the official
NMVOC and $CO_2$ relative changes reported for the industrial manufacturing sector, respectively. Oppositely, for doumbiaetal
and forsteretal large discrepancies are observed with official estimates, especially in the case of NMVOC, where the reductions
reported at the EU27+UK level are 4.5 and 8.5 times larger, respectively. Both guevaraetal and liuetal consider the use of
industrial production indexes as a proxy for this sector, while doumbiaetal and forsteretal rely on Google mobility data.

It is worth noting how guevaraetal reproduces the heterogenous changes across both pollutants at the EU27+UK level, with
NMVOC presenting an approximately 2 times lower reduction (-2.9%) than $CO_2$ (-6.7%). This result can be partially explained
by the fact that during the lockdowns the food and chemical industries (both of which contribute significantly to total NMVOC
industrial emissions) were considered to be essential; as a consequence, their activity was less reduced than that of other
energy-intensive industrial branches such as iron and steel manufacturing or non-metallic mineral products, which present
larger contributions to total industrial $CO_2$ emissions.

Inconsistencies between emep_ceip and unfccc official emission changes are observed for Cyprus and Malta, with NMVOC
emissions remaining almost unchanged while $CO_2$ show increases by 20% to 30%. In the case of a small countries like these
two, the national emissions are rather sensitive to dynamics at the single facility level, resulting in large relative year to year
changes.

### 3.1.3    Residential/commercial stationary combustion activities

For this sector, relative PM2.5 emission changes reported by all the near-real time datasets are inconsistent with official
estimates (Fig. 2e and 2f). While the first group indicates a general increase of emissions, the former reports a decrease in
almost all European countries. The differences between official and non-official estimates are in general much larger for
doumbiaetal and forsteretal than for guevaraetal. This could be explained by the fact that, while all three datasets use Google
mobility reports as a data proxy for this sector, guevaraetal is the only one that adjusted the original values considering energy
consumption statistics from the residential and commercial sectors (Guevara et al., 2022). All in all, the message from the
three near-real time estimates is the same: during 2020 people spent more time at home due to confinement measures and
therefore the consumption of residential wood combustion, which represents more than 90% of total PM2.5 from this sector,
increased when compared to a 2020 business-as-usual situation. Nevertheless, and as explained in Sect. 2.3, relative changes
reported by emep_ceip use 2019 as a baseline, and therefore they include the effect not only of COVID-19, but also the impact
of meteorological changes. As reported by the Copernicus Climate Change Service (C3S), Europe experienced its warmest
winter on record in 2020, with temperatures up to 5°C warmer than the 1981–2010 seasonal average in north-eastern Europe
(C3S, 2020a). The impact of the exceptionally mild winter temperatures in 2020 is illustrated in Fig. 4a, which shows the





relative changes in number of Heating Degree Days (HDD) per European country between 2019 and 2020 (Eurostat, 2022).
       Overall, HDD was -5% lower in 2020 when compared to 2019, with values up to -20% in Malta and approximately -10% in
       Finland, France or Estonia. However, the increase of temperatures was not uniform and some countries such as Bulgaria and
       Hungary presented increases in the HDD of approximately +5%. Because HDD is an indicator designed to describe the energy
       requirements of buildings, a decrease (increase) in its value implies a decrease (increase) in the combustion of fuels and
associated emissions needed for space heating. Figure 4.b shows a scatterplot of relative changes in PM2.5 emissions as
       reported by emep_ceip and in HDD per country. It is observed in several countries that a clear relationship is identified, with
       emissions decreasing when the HDD decreases (e.g., Finland) and the other way around (e.g., Bulgaria). These findings are
       consistent with those reported by Ciais et al., (2022) using ENSTO-G daily gas consumption data in buildings, who also
       showed that climate variations played a larger role in residential energy consumption across Europe in 2020 than COVID-19
induced stay-home orders, except in Italy and France. Nevertheless, the relationship is not always consistent. For instance, in
       Estonia PM2.5 emissions and HDD present relative changes of similar magnitude but of opposite sign (+ 10% and -10%,
       respectively), indicating that other factors, such as fuel-switching or inconsistencies in the officially reported emission time
       series, among others, could play a role.

For $CO_2$, it is observed that liuetal is the near-real time estimate that is generally more in line with the official unfccc emission
       changes. This result is consistent with the fact that of the near-real time datasets only liuetal accounts for the impact of
       meteorology, which at the same time reinforces the hypothesis that changes in this sector are mainly driven by changes in the
       meteorology. As a matter of fact, liuetal assumes that changes in emissions are only driven by changes in population weighted
       2m temperature for this sector, and no impact from COVID-19 is included in the 2020 emissions. This can be illustrated by
the fact that both liuetal and the relative changes in the HDD point out Malta as the country experiencing the largest decrease
       (around -20% in both cases). This result, however, contrasts with the relative changes reported by unfccc, which indicate a
       nearly +10% increase in $CO_2$ emissions in this country.

### 3.1.4    Fugitive emissions from fossil fuels

Fig. 5a and 5b shows the relative changes in $CH_4$ emissions from activities related to the extraction, processing and delivery
of fossil fuels to the point of final use. Guevaraetal is the only near-real time dataset that reports information for this sector
       while all the other estimates assume no changes in emissions during 2020. Results are fairly in line with official estimates at
       the EU27+UK level (-5.2% versus -7.2% for $CH_4$). It is also interesting to see how guevaraetal can reproduce the large drop
       in emissions occurred in Greece (close to -40%), which is related to a significant decrease in coal mining activities (Guevara
       et al., 2022).


The official reported drop in European $CH_4$ emissions (not only for this sector but also for total emissions as stated in Table
4) contrasts with recent observational-based studies that claimed increases in $CH_4$ emissions during 2020 using TROPOMI





observations and inverse-modelling techniques (McNorton et al., 2022). As reported by Stevenson et al. (2022), the increased $CH_4$ atmospheric growth captured by TROPOMI is probably due to the net effect of NOx, CO, and NMVOC emission changes

on $CH_4$ atmospheric lifetime rather than on changes in primary emission sources.

### 3.1.5    Solvents

For the sector solvent use, only guevaraetal reports changes in emissions in 2020, as the other near-real time datasets do not report information for this sector. However, the changes estimated by guevaraetal only focus on a few industrial activities (i.e., metal degreasing and printing) and do not cover the domestic use of solvents, which results in a very limited change of the

total NMVOCs at the EU27+UK level (-1.3%). Interestingly, large inconsistencies are observed in the official relative changes reported between European countries. While most of them indicate changes in total emissions between -5% and +5%, significant increases (e.g., +50% in the Netherlands, +33% in Finland, +25% in Portugal) and decreases (e.g., -25% in Lithuania) are observed in certain countries. This inconsistency is mainly driven by the heterogeneous estimation of changes in NMVOC emissions from the use of the so-called pandemic products (e.g., hand sanitizer gels). This hypothesis is illustrated

in Fig. 3b, which shows official relative NMVOC emission changes for the domestic solvent use sector (NFR2D3a) are shown. Similarly to what is observed at the GNFR level, NMVOC emission changes from this activity are very heterogeneous across countries, with Portugal, the Netherlands and Finland presenting increases larger than 100%, and many other presenting changes ranging from -5% to 5%. The COVID-19 recommendation on the use of hand sanitizers as a safety measure was a measure consistently implemented across European governments during 2020 and, therefore, its impact on NMVOC emissions

from this activity should be, in theory, also consistent across national reported inventories. However, several countries use a very basic emission estimation method (tier 1) for this activity, which uses population data as activity data and thus does not reflect the increased use of hand sanitizers.

### 3.1.6    Road transport

For the traffic sector, guevaraetal is the dataset more in line with the NOx and $CO_2$ relative changes reported by official

estimates at EU27+UK level and in those countries that were most affected by COVID-19 restrictions (e.g., Spain, Italy, UK, France) (Fig. 5c). Doumbiaetal and forsteretal present the largest discrepancies with official estimates, with relative changes between 1.5 and 2 times larger on average. When compared to unfccc, results by liuetal tend to present lower emission changes (2.5 times lower at the EU27+UK level). Differences are particularly relevant in those countries where liuetal suggests almost no changes in emissions (e.g., Austria, Germany) or even slight increases (e.g., Estonia, Lithuania). In fact, official estimates

do not report any country with increasing road-transport emissions in 2020, Romania being the country closest to a negligible change (-2.3% for $CO_2$). Out of all the near-real time estimates, guevaraetal and liuetal are the only ones that combined the use of new mobility metrics (Google reports and Tomtom congestion statistics, respectively) with traditional statistics (measured traffic counts) to derive the impact of COVID-19 restrictions on this sector. Contrary to what is shown for the public



power or the residential sectors, the good agreement observed between guevaraetal and ceip_emep/unfccc suggests that the
changes in emissions from this sector where almost exclusively related to the COVID-19 mobility restrictions.

### 3.1.7    Aviation

The aviation sector reports the largest drops in emissions according to all official and non-official estimates (Fig. 5.e). At the
same time, it is also the sector with the fewest differences in estimates at the EU27+UK level, with overall $CO_2$ reductions
ranging from -53% to -58%, except for doumbiaetal, which report much lower reductions (-41.7%). The analysis at the country
level though suggests that liuetal is the dataset that is more in line with the official results reported by unfccc. The reduction
of emissions is quite consistent across countries, except for Bulgaria and Luxemburg, where reductions are significantly below
the average (-30% and -10%, respectively) and only liuetal is capable of partially reproducing them (-40% and -20%,
respectively). Results by doumbiaetal tend to underestimate the reductions reported by unfccc by a factor of 1.6 on average
(e.g., -66% versus -54% for Italy and -57% versus -37% for Poland). While unfccc and liuetal reports changes in emissions
from landing and take-off (LTO) and cruise domestic operations, doumbiaetal, guevaraetal and forstersetal only reflect changes
from LTO from both domestic and international air traffic, which could explain why the discrepancies are larger.

### 3.1.8    Off-road mobile sources

As for the case of the fugitive emissions from fossil fuels (Sect. 3.1.4), for off-road mobile source emissions only guevaraetal
considers the impact of COVID-19 restrictions. However, and as shown in Fig. 5.f, significant discrepancies exist between
this dataset and the official emep_ceip estimates, with the former reporting larger $NO_x$ emission reductions (-7.3% versus -
1.7%). The methodology of guevaraetal considered the impact of the mobility restrictions only in industrial machinery,
assuming that other types of machinery included in this sector (i.e., agricultural, gardening, recreational boats) were not
affected by the pandemic. Interestingly, all official estimates report a decrease in emissions, except for the cases of Portugal
and Greece, where increases of +6% and +38% are observed, respectively.






**Figure 2** Relative emission changes [%] reported by official (emep_ceip, unfccc) and non-official datasets (guevaraetal, forsteretal, doumbiaetal, liuetal) per country and at the EU27+UK level for the public power sector (a,b), manufacturing industry sector (c, d) and other stationary combustion activities (e, f)




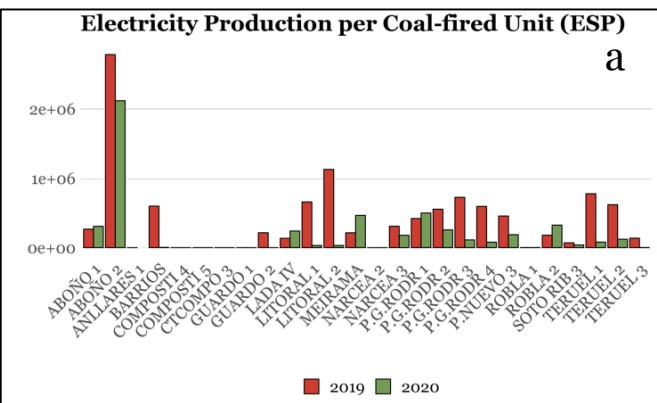

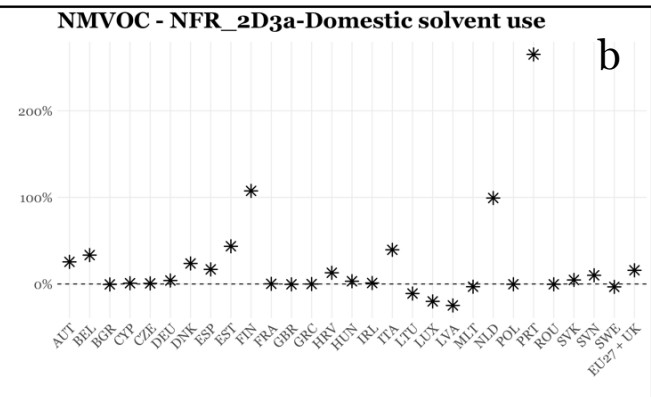

**Figure 3 Annual electricity production reported per individual coal-fired power plant in Spain during 2019 and 2020 (ENTSO-E, 2022) (a) and relative emission changes [%] reported by official estimates (emep_ceip) per country and at the EU27+UK level for the domestic use of solvent activity (b)**





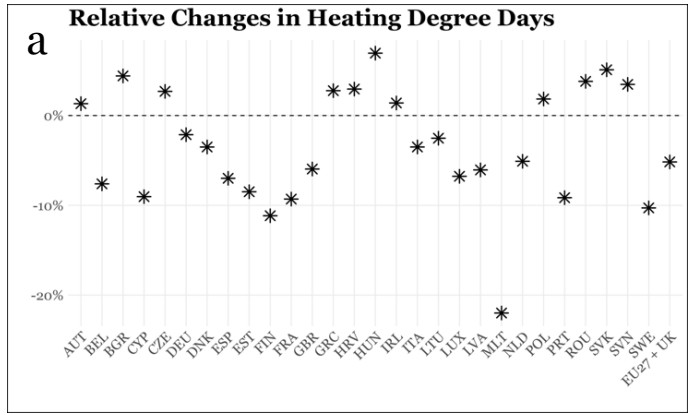

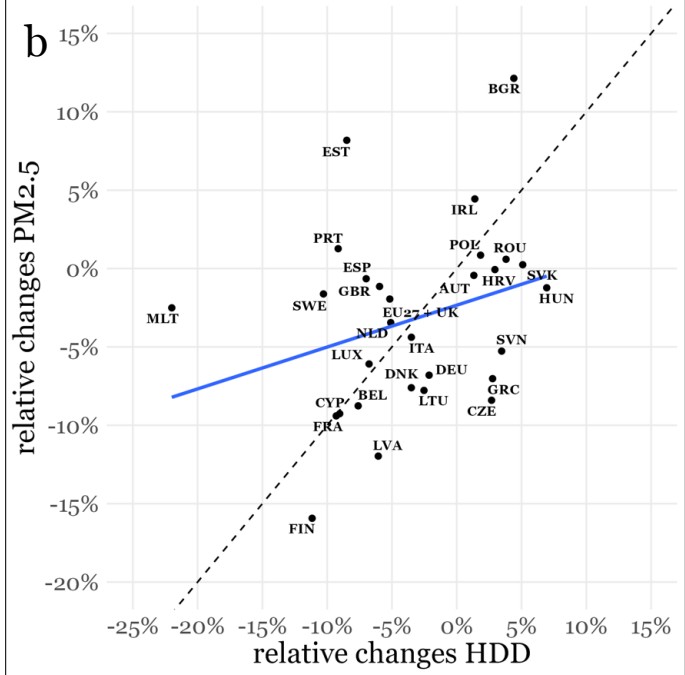

**Figure 4 Relative changes [%] in the number of Heating Degree Days (HDD) per country and at the EU27+UK level between 2019 and 2020 (Eurostat, 2022) (a) and scatter plot showing the relative changes in the HDD and in PM2.5 emissions from the residential/commercial sector per country and at the EU27+UK level (b).**








**Figure 5 Relative emission changes [%] reported by official (emep_ceip, unfccc) and non-official datasets (guevaraetal, forsteretal, doumbiaetal, liuetal) per country and at the EU27+UK level for the fugitive fossil fuel sector (a), use of solvents sector (b) road transport (c, d), aviation (e) and the off-road mobile sources (f)**





## 3.2 Monthly and quarterly emission changes

Figure 6 shows the relative changes in monthly $NO_x$ and $CO_2$ emissions occurred in the EU27 + UK as reported by each of the
near-real time datasets described in Sect. 2.2. Official reported data could not be included in this comparison as emissions are
available only at the annual level for most of the countries, and just a few of them publicly disclose information at a finer
resolution (i.e., monthly, quarterly), as discussed later in this section.

For total $NO_x$ and $CO_2$, a similar temporal pattern is reported by the four datasets, with: (i) the largest drops occurring during
the first round of lockdowns (March to May), (ii) emissions getting closer to pre-pandemic levels when national governments
rolled back COVID-19 measures (June to September) and (iii) a new round of lower intensity drops associated with the second
pandemic wave in Europe (October to December). However, discrepancies exist regarding the magnitude of the changes
reported by each dataset over the three periods.

For $NO_x$, the drops reported by guevaraetal during March-May and October-December are 1.3 to 2.3 times lower than those
provided by forsteretal and doumbiaetal. Significant differences of a similar magnitude are also observed during summertime,
when doumbiaetal and forsteretal report much larger reductions when compared to guevaraetal. These discrepancies are mainly
driven by the different $NO_x$ emission changes estimated for road transport during the same periods (Fig. 6c). When looking at
the $NO_x$ emissions changes in the manufacturing industry sector (Fig. 6d), discrepancies between datasets occur both in terms
of the magnitude and timing of the drops. Concerning the temporal aspect, both doumbiaetal and forsteretal reproduce a pattern
similar to that of road transport emissions, with a first drop occurring during March-May (reductions up to -53% and -55% in
April), a recovery period during the summer and a second drop between November and December (reductions up to between
-29% and -32% in December). Oppositely, guevaraetal results suggest a pronounced recovery from May onwards, with
emission reductions reaching levels very close to business-as-usual by the end of the year (-0.05% in December). These results
are in line with the fact that most restrictions imposed in October, November and December were generally slower and softer
than those implemented in March-April (e.g., curfews, limited social gatherings, early closing times for restaurants and bars),
and had no effect on the manufacturing industry. The differences between doumbiaetal/forsteretal and guevaraetal results can
be directly linked to the activity proxies considered for the manufacturing industrial sector. The first two datasets considered
Google mobility data to estimate changes in industrial emissions, whereas guevaraetal results are based on changes in industrial
production indices.

For road transport $CO_2$ emissions (Fig. 6d), the drops reported by liuetal in April (around -28%) are almost 2 times lower than
those estimated by the other three datasets (between -50% and -60%). For this sector, the consistency observed between
guevaraetal, doumbiaetal and forsteretal during the first wave of the COVID-19 epidemic (i.e., March, April and May) is
dissipated in summer, specially during July and August, when forsteretal suggests important decreases in emissions (close to



-20%), doumbiaetal indicates reductions around -10% and guevaraetal reports moderate decreases (approximately -5%). The drops reported for traffic $CO_2$ emissions by forsteretal and doumbiaetal are back in line during the second wave of contamination (i.e., November and December, close to -40%), with the results estimated by guevaraetal and liuetal being much lower once again (between 2 and 5 times). For $CO_2$ emissions from the public power sector (Fig. 6f), liuetal already reports

significant drops in January and February (approximately -20%), before the beginning of the pandemic. This result reinforces the hypothesis discussed in Sect. 3.1.1, which indicates that changes in 2020 emissions from this sector were mainly driven by national coal phase out commitments that have been continuously implemented since the UN Paris Agreement was adopted during the COP21 in December 2015. For this sector, results reported by guevaraetal and doumbiaetal are generally in line, since in both cases the electricity demand data from ENTSO-E is used as the main proxy to derive the emission adjustment

factors (Table 2).





**Figure 6 Relative NOₓ and CO₂ monthly emission changes [%] reported by each near-real time dataset at the EU27+UK level for total emissions except shipping (a, b) and selected sectors including road transport (c, d), manufacturing industry (e) and public power (f).**




Figures 7 and 8 present a comparison of the near-real time estimates against publicly disclosed national monthly (France, CITEPA, 2022) and quarterly (UK, BEIS, 2022; the Netherlands, CBS, 2022) estimates reported by national inventory agencies. For UK and the Netherlands, official results are only provided for GHGs and 5 general sectors, whereas for France information is available for both AP and GHGs at a detailed activity level (75 subsectors), allowing a more extended
comparison (i.e., $NO_x$ and $CO_2$ for total emissions and selected sectors).

The guevaraetal results are the ones closer to the French $NO_x$ official estimates (i.e., CITEPA) during the periods corresponding to the two main waves of pandemic prevention and control policies (i.e., March-May and October-December). This consistency is observed for total emissions (Fig. 7.a) as well as for the road transport (Fig. 7.c) and industrial manufacturing (Fig. 7.e)
sectors. The largest discrepancy between the two datasets is observed in April (-49% versus -38%) and is mainly driven by differences in the manufacturing industry sector (-38% versus -26%), the results reported for road transport being the same (-64%). The doumbiaetal and forsteretal datasets tend to overestimate the official $NO_x$ emission reductions during the two lockdown periods, the largest discrepancy occurring for the manufacturing industry sector in November and December, when the two near-real time datasets indicate reductions of around -30% while CITEPA reports values above BAU levels (up to
9%). This inconsistency is in line with the results from Fig. 6.e previously discussed. The drops of total $NO_x$ emissions occurred during April and May (-38% and -27%) are also overestimated by both doumbiaetal (-59% and -47%) and forsteretal (-60% and -45%). Regarding total $CO_2$ emissions (Fig. 7.b), guevaraetal and liuetal are in general the datasets more in line with official estimates. The same conclusion is obtained when looking at the results for the road transport sector (Fig. 7.d). The drops reported by CITEPA during April and May (-63% and -37%) are well reproduced by guevaraetal (-61% and -33%),
slightly underestimated by liuetal (-50% and -26%) and significantly overestimated by doumbiaetal (-80% and -59%) and forsteretal (-79% and -57%). As shown before (Sect. 3.1.1), liuetal is the dataset that generally reproduces better the official changes reported for the public power sector (Fig. 7.f), being able to capture the increases occurred during summertime, which are partially linked to the record temperatures experienced in France (C3S, 2020b) and the associated increase in the energy demand for the use of air conditioning systems. Despite the good agreement between liuetal and CITEPA for this sector, some
important discrepancies are still observed mainly in April, when the near-real time dataset significantly overestimates the reported drop (-44% versus -71%).

The official relative $CO_2$ quarterly emission changes estimated by BEIS for the UK are in good agreement with the results reported by guevaraetal and liuetal, while a general overestimation is observed for doumbiaetal and forsteretal (Fig. 8.a). All
datasets report the largest drop in the second quarter of the year, i.e., -24% according to BEIS and guevaraetal, -30% according to liuetal, -33% according to doumbiaetal and -35% according to forsteretal. For the Netherlands, liuetal is the one closer to the CBS official estimates for all quarters (e.g., -15% in both cases during the second quarter), the results by forsteretal and doumbiaetal being again the ones that present the largest discrepancy (Fig. 8.b). Interestingly, the drop of $CO_2$ emissions reported during the first quarter of the year (-11%) is of the same magnitude as the ones reported during the second (-15%)



and fourth (-10%) quarters, when national lockdowns were implemented. This drop is only partially reproduced by liuetal and it is mainly related to a drop in the $CO_2$ emissions from the power sector (not shown), which was triggered by the retirement of hard coal-fired power plants by the end of 2019.

**Figure 7 Relative NOₓ and CO₂ monthly emission changes [%] reported by each near-real time dataset and CITEPA (2022) for**
**France for total emissions except shipping (a, b) and selected sectors including road transport (c, d), manufacturing industry (e) and public power (f).**





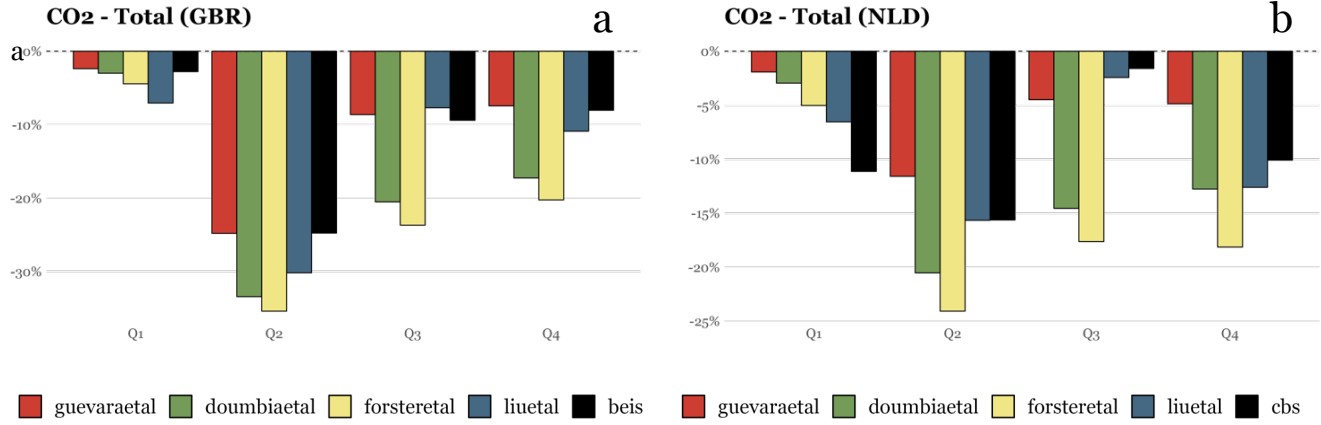

**Figure 8 Relative total $CO_2$ quarterly emission changes [%] reported by each near-real time dataset and official estimates from BEIS (2022) and CBS (2022) for UK (a) and the Netherlands (b), respectively.**



## 4 Conclusions

This work presents the results of an intercomparison of relative European anthropogenic emission changes in 2020 reported by official and non-official estimates. Official estimates include the national inventories of air pollutants (AP; $NO_x$, NMVOC,
$SO_2$, $NH_3$, PM2.5) and greenhouse gases (GHG; $CO_2$ and $CH_4$) reported under the CLRTAP and the UNFCCC, respectively. The selection of near-real time emission estimates includes the CAMS COVID-19 European emission adjustment factors (guevaraetal), the global CONFORM dataset (doumbiaetal), the COVID-19 estimates developed by Forster et al. (2020) (forsteretal) and the $CO_2$ emission estimates reported by the Carbon Monitor initiative (liuetal). The comparison focusses on the EU27 + UK and is performed on an annual, quarterly and monthly basis. The following conclusions were obtained from
the intercomparison work:

- $NO_x$ and $CO_2$ are consistently being reported by official and non-official estimates as the pollutants that experienced the largest reductions in Europe in 2020 (-11.3% and -12.2% according to official estimates). Similarly, $NH_3$ and $CH_4$ are reported by official and the near-real time datasets as the species with the lowest reductions (i.e., -1.1% and -1.4% according to official estimates and between -0.9% and 0.1% according to non-official estimates).

- Despite this agreement, large discrepancies arise between the official and non-official datasets when comparing results for specific sectors and countries.

- The guevaraetal dataset tends to be more in line with official AP relative emission change estimates, while the results reported by forsteretal and doumbiaetal, which are largely derived from Google mobility data, present larger discrepancies.

- Results reported by liuetal are generally in a good agreement with official $CO_2$ estimates, except for the road transport sector, where they tend to report relative emission reductions much lower than those provided by the UNFCCC official inventories.

- For the residential combustion, public energy industry and shipping sectors, changes in emissions occurred between 2019 and 2020 were mainly dominated by non-COVID-19 factors, such as meteorology (i.e., warmer winter), the
implementation of national decarbonization plans in the electricity sector, and the introduction of the Global Sulphur Cap rule, respectively.

- The increase in NMVOC emissions from the use of pandemic products (e.g., hand sanitizer gels) is heterogeneously considered in official CLRTAP inventories, as several countries use a very basic emission estimation method (tier 1) that uses population data as activity data and thus does not reflect the increased use of these products.

- Relative changes in AP and GHG emissions reported by the CLRTAP and UNFCCC official estimates are in general consistent. However, some discrepancies were detected in some cases (e.g., changes in $SO_2$ versus $CO_2$ emissions from public power), which could be attributed to issues with the reported data or the coordination between AP and GHG inventory development efforts.





- Regarding monthly relative changes in total $NO_x$ and $CO_2$, similar patterns are observed in the different near-real time estimates, with the largest drops occurring during first round of lockdowns (March to May), emissions getting closer to business-as-usual levels between June and September, coinciding with the ease of restrictions, and a new round of lower intensity drops occurring between October to December, when a second pandemic wave affected Europe. However, important discrepancies exist regarding the magnitude of the changes reported by each dataset during the three periods, which are again related to the different activity proxies used to estimate the drops in emissions.

- When compared to official quarterly and monthly estimates reported by national inventory agencies, guevaraetal and liuetal are again the datasets that are in a better agreement, both for total emissions and specific sectors, including road transport, manufacturing industry and public power.

- The present intercomparison work does not allow checking the quality of the near-real time estimates in an absolute way since, even being based on local data and detailed estimation methodologies, official national emission inventories have also uncertainties associated to them and cannot be considered as the ground truth. Nonetheless, the cases where datasets converge on similar trends could be interpreted as providing an encouraging cross-verification of the official and independent emission inventories.

The COVID-19 outbreak has remarkably contributed to a crucial change in how we quantify and understand emissions of AP and GHG. New datasets and proxies based on inter alia mobility and congestion data derived from smartphones or GPS systems have emerged that did not exist before or were not extensively being considered by the emission modelling community. The near-real time estimates presented in this work demonstrate how emission compilation methodologies can take advantage from the emergence of big data from remote sensing technologies and smart devices. The irruption of these technologies and associated datasets, which are expected to continue growing, provides the opportunity for a change of paradigm in the production of emission estimates for monitoring and modelling applications, mainly air quality forecasting. As proposed by Tong et al. (2012), improved predictions of air quality require bringing emission science to a new level and moving from inventory-based data processing approaches (i.e., generation of hourly model ready emission data by processing existing and pre-calculated annual emission estimates) to modelling approaches that use and integrate near-real time data collected from multiple networks and monitors. The need for near-real time emission information has grown not only because of the COVID-19 pandemic, but also as a result of an increased interest from the general public in climate mitigation and environmental protection, as well as subsequent events that are causing disruptions to the business-as-usual emission levels, most notably the war in Ukraine and the associated energy crisis.

Despite the new opportunities created by the aforementioned technological advancements, estimating emissions in near-real time still presents several challenges. Firstly, the results of this intercomparison work highlight that caution is required when using new mobility data to estimate changes in emissions, and that these proxies should be combined with traditional statistics such as measured traffic counts or energy consumption statistics. However, traditional information is still difficult to be





acquired in a consistent way, particularly when working at the global level. As previously highlighted, the results reported by the guevaraetal dataset, which covers only Europe, are generally more in line with official estimates. This demonstrates how difficult is to obtain accurate and consistent local information when working at the global level. At the same time, differences between guevaraetal and official reported emissions for specific sectors and countries indicate that uncertainties are large, even in case of large disturbances such as the COVID-19 pandemic, and that current approaches might miss normal interannual variations. Secondly, digitized near-real time information arising from new smart technologies covering key sectors such as electricity production, aviation or road transport is emerging; however, for some other relevant activities, such as use of solvents, residential and commercial combustion (particularly residential wood combustion) and agricultural activities, it is likely that near-real time activity monitoring will remain scarce. Observations from satellite-based sensors are key to partially overcome this limitation, as exemplified by the Global Fire Assimilation System (GFAS; Kaiser et al., 2021) for monitoring biomass burning emissions or the use of very high-resolution satellites (e.g., WorldView3, WV3) to detect and quantify $CH_4$ emitters (Irakulis-Loitxate et al., 2022), among others.

## 5    Data availability

Officially AP (i.e., $NO_x$, NMVOC, $SO_2$, CO, $NH_3$ and PM2.5) and GHGs (i.e., $CO_2$ and $CH_4$) reported emissions for 2019 and 2020 (reporting year 2022) were obtained from https://www.ceip.at/webdab-emission-database/reported-emissiondata and https://unfccc.int/ghg-inventories-annex-i-parties/2022, respectively. The collection of the CAMS COVID-19 emission adjustment factors reported by Guevara et al. (2022) are available from https://doi.org/10.24380/k966-3957. The CONFORM emission adjustment factors reported by Doumbia et al. (2021) are available from https://permalink.aeris-data.fr/CONFORM. The emission adjustment factors reported by Forster et al. (2020) are available from https://github.com/Priestley-Centre/COVID19_emissions. The $CO_2$ European emissions reported by Carbon Monitor (Liu et al., 2020) are available from https://eu.carbonmonitor.org/. Copernicus CAMS-REG_v5.1 2020 business-as-usual (BAU) emission inventory (Kuenen et al., 2022b) is distributed from the Emissions of atmospheric Compounds and Compilation on Ancillary Data (ECCAD) system (https://doi.org/10.24380/eptm-kn40).

## 6    Authors contributions

MG conceived and coordinated the study. MG, HP, OJ and CPGP prepared the comparison plots and contribute to the interpretation and discussion of the results. HACDvdG, JK, IS, CG, TD, PC, ZL, SS, MB and RL contributed to the interpretation and discussion of the results. OJ and CPGP supervised the work. MG prepared the manuscript with contributions from all co-authors.



## 7 Competing interests

The authors declare that they have no conflict of interest.

## 8 Acknowledgments

The research leading to these results has received funding from the Copernicus Atmosphere Monitoring Service (CAMS),
620 which is implemented by the European Centre for Medium-Range Weather Forecasts (ECMWF) on behalf of the European
Commission. We acknowledge support from the VITALISE project (PID2019-108086RA-I00) funded by
MCIN/AEI/10.13039/501100011033; from the MITIGATE project (PID2020-116324RA695
I00/AEI/10.13039/501100011033) from the Agencia Estatal de Investigación (AEI); from the BROWNING project (RTI2018-
099894-BI00) from the Ministerio de Ciencia, Innovación y Universidades; from the AXA Research Fund; and from the
625 European Research Council (grant no. 773051, FRAGMENT).



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
