# Peer review of "Towards near-real time air pollutant and greenhouse gas emissions: lessons learned from multiple estimates during the COVID-19 Pandemic"

_EGUsphere, 2023_

## Author Comment (AC1)

We would like to thank the two reviewers for their positive and constructive feedback, which helped improve the quality of the paper The review comments have been helpful in pointing out parts that required further improvements. Below we address specific issues mentioned by the reviewers point by point. The manuscript has been updated accordingly.

**Anonymous Referee #1**

This is an interesting piece of work comparing European emission inventories for the year 2020 and how they reflect the impact of the emission reductions caused by lockdown measures taken to prevent the spread of the COVID-19 virus. The idea behind the analysis is that lessons can be learned about the construction of near-real-time emission inventories that will be able to reflect large unexpected short-term changes in emissions. This is a very valuable exercise and the conclusions that the authors draw are worth to be published and brought to a wider audience. I recommend publication of this work after few smaller modifications.

Detailed comments:

Line 32: between -29.3 and -5.6 %

Corrected

Line 73: give examples which countries reported quarterly and monthly emission estimates.

Examples have been added as follows:

"(···) some European countries started to publish quarterly (e.g., UK, BEIS, 2022; the Netherlands, CBS, 2022) and monthly (e.g., France; CITEPA, 2022) estimates of emissions based on preliminary energy data" (line 74 of the revised manuscript)

Line 151 -154: reformulate this sentence. It can be read as if you use shipping mobility statistics for aviation emission. It remains also unclear what exactly shipping mobility statistics is and where the data can be found.

The sentence has been reformulated to clarify better which proxy is used for each emission sector. We have also updated and provided more details on the statistics considered for the shipping sector (both in the text and in Table 2):

"The activity proxies considered include electrical power generation for power plant emissions, production data and production indices of industry processes for industrial manufacturing emissions, mobility indices (TomTom data for > 200 cities in Europe aggregated to country scale) for road transport emissions, flight location data (FlightRadar24 database) for aviation emissions and shipping mobility statistics (metric tons of cargo from the UN COMTRADE Monitor database) for maritime emissions" (lines 155 to 159 of the revised manuscript)

Table 2: For Forster et al the emission changes in aviation and shipping are given to be based on a previous publication by Le Quéré et al. Could you say how they derived their adjustment factors for these sectors?

A description on how the Le Quéré et al. emission changes in aviation and shipping were derived has been added as follows:

"For the air traffic and maritime sector, the Le Quéré et al. (2020) emission trends for international and national aviation and shipping are directly used, which were derived from total number of departing flights reported by the Official Aviation Guide of the Airway (OAG) and shipping activity forecasts provided by the World Trade Organization, respectively" (lines 144 to 145 of the revised manuscript)

Line 189 – 191: It would be nice if you could say more about the differences for the shipping sector. If some authors report only global changes, this means that they apply the same reductions in Europe and you should be able to compare those to the factors calculated with the STEAM model.

Following the reviewer's suggestion, we completed Tables 3 and 4 by including the relative changes in emissions for the shipping sector at the EU27+UK level as reported by doumbiaetal, forsteretal and liuetal. The discussion of these results was also added in Section 3.1 of the manuscript. As expected, important discrepancies are observed between these results and the official estimates, as doumbiaetal, forsteretal and liuetal report only global emission changes for this sector, and do not distinguish between European and non-European seas. For instance, if we focus on $CO_2$ emissions, we can see that the relative reductions reported by liuetal (-3.1%) and doumbiaetal (-9.5%) are lower than official UNFCCC estimates (-11%). The largest discrepancy is reported by forsteretal (-23.5%), which could be related not only to the differences in terms of spatial coverage, but also to the fact that for this database emission trends for shipping were derived from forecasted activity (provided by the World Trade Organization) rather than measured statistics.

Figure 1: Would be good to include GNFR sector names in addition to the letters, similar to Tables 3 and 4.

The figure has been updated following the reviewer's recommendation.

Line 262: omit one "slightly"

Corrected

Line 373 – 385, section 3.1.6 Road transport: $CO_2$ from road transport is shown in Figure 5d, however, this sub-figure is never mentioned in the text, and its contents is only touched upon. You might write a bit more about Figure 5d and also make clearer when you talk about NOx and when about $CO_2$.

We have re-written this section to better clarify when we are referring to NOx or $CO_2$ emission results. We have also added references to Figure 5d.

Line 415: Figure 3a and 3b don't fit together into one. They show very different things and are mentioned at very different places in the text.

Authors agree with the reviewer. We decided to move Figure 3a to the supplementary material - as it is not fundamental to have it in the main text - and keep only Figure 3b. The text where the figures are mentioned and the figure's captions have been revised accordingly.

Line 589/590: how difficult it is

The sentence has been reformulated to clarify better the challenge:

"However, this traditional information is still difficult to be acquired in a consistent and homogeneous way, particularly when working at the global level, as the number of global repositories giving access to near-real time and high-resolution emission proxy information is very scarce" (lines 634 to 635 of the revised manuscript)

References: give last access date for Kinsey (2020)

We replaced this reference by another one that better reflects the dataset currently used by CarbonMonitor to estimate the daily variability of shipping emissions. Table 2 of the manuscript has been revised accordingly.

Cerdeiro, D.A., Komaromi, A., Liu, Y., Saeed, M.: World Seaborne Trade in Real Time: A Proof of Concept for Building AIS-based Nowcasts from Scratch. IMF Working Paper. Working Paper No. 2020/057, available at: https://www.imf.org/en/Publications/WP/Issues/2020/05/14/World-Seaborne-Trade-in-Real-Time-A-Proof-of-Concept-for-Building-AIS-based-Nowcasts-from-49393 (last access: May 2023), 2020.

**Anonymous Referee #2**

The paper "Towards near-real time air pollutant and greenhouse gas emissions: lessons learned from multiple estimates during the COVID-19 Pandemic" by Guevara et al. provides a very useful comparison of several "real time" estimates and official inventory data.

The paper is well written and provides a very useful and informative analysis. I have only two comments.

One minor point that should be added is that the official inventory estimates for 2020 are also approximations of actual emissions. There are two components of this. Of course all inventories may contain errors. However, more relevant for this comparison, is that inventory estimates for the last year published often change in subsequent revisions as underlying data (such as energy statistics) and other information is updated or revised. I suspect this won't impact the comparisons for the most part, but occasionally (for some country/sector combinations) this might have a signifiant impact.

We included the point suggested by the reviewer in the conclusions section as follows:

"•	Linked to the previous point, official emission inventory estimates are subject to continuous revisions as the underlying data (e.g., energy statistic, emission factors) and estimation methodologies are updated or improved every year. These revisions may occasionally incur significant changes to emissions from specific countries/sectors/species (e.g., Kuenen et al., 2022), and subsequently to the corresponding comparison results presented in this work." (lines 614 to 617 of the revised manuscript)

The collection of these various inventory data took some signifiant work and, while all the data are publicly available, it would be laborious for someone to replicate the work done by the authors. The authors should, as a supplmental data file, supply the following information: annual emissions and/or adjustment factors (as appropriate) by country/sector/year, and the same data by month (where available). This would make the analysis in this paper more readily useful to the community since the comparisons could then easily be extract for any country/sector of interest for use in subsequent analysis.

We constructed a numeric file that contains annual and monthly emissions per country, GNFR sector and pollutant. For the official inventories (EMEP-CEIP and UNFCCC) and Liu et al. (2020) we provide the corresponding emissions reported for the years 2019 and 2020. For Guevara et al., (2022), Doumbia et al., (2021) and Forster et al., (2020) we provide, on the one hand, the CAMS-REG_v5.1 business-as-usual (BAU) 2020 emissions and, on the other hand, the results of combining this inventory with the adjustment factors reported by each one of the three databases. The file is provided in excel format as part of the supplementary material of the paper, and it includes a README sheet describing each one of the information fields. The file is mentioned and briefly described in the "Data availability" section of the paper (lines 662 to 667 of the revised manuscript).